# Effect of Calcination Temperature on the Microstructure, Composition and Properties of Agglomerated Nanometer CeO_2_-Y_2_O_3_-ZrO_2_ Powders for Plasma Spray–Physical Vapor Deposition (PS-PVD) and Coatings Thereof

**DOI:** 10.3390/nano14120995

**Published:** 2024-06-07

**Authors:** Zhenning Hou, Wenchao Yang, Yongzhong Zhan, Xiaofeng Zhang, Jingqin Zhang

**Affiliations:** 1State Key Laboratory of Featured Metal Materials and Life-Cycle Safety for Composite Structures, MOE Key Laboratory of New Processing Technology for Nonferrous Metals and Materials, School of Resources, Environment and Materials, Guangxi University, Nanning 530004, China; 17318501424@163.com (Z.H.); zjq18839663999@163.com (J.Z.); 2The Key Laboratory of Guangdong for Modern Surface Engineering Technology, National Engineering Laboratory for Modern Materials Surface Engineering Technology, Institute of New Materials, Guangdong Academy of Sciences, Guangzhou 510650, China

**Keywords:** thermal barrier coating, CeO_2_-Y_2_O_3_-ZrO_2_ (CYSZ), doping mechanism, high-temperature performance, thermal spray technology

## Abstract

Self-made agglomerated nanometer CeO_2_-Y_2_O_3_-ZrO_2_ (CYSZ) powders for plasma spray–physical vapor deposition (PS-PVD) were prepared by spray-drying, followed by calcination treatment at four different temperatures (600 °C, 700 °C, 800 °C, 900 °C). The physical properties, microstructure, and phase composition of the calcined powders were investigated using a laser particle size analyzer, scanning electron microscopy (SEM), and X-ray diffraction (XRD). The results showed that compared to the agglomerated powders obtained through spray-drying, the particle size of the agglomerated powders changed with increasing calcination temperature, accompanied by an increase in the self-bonding force of the agglomerated powder particles. The proper calcination temperature improved the sprayability of the powders. Additionally, with the increase in the calcination temperature, a transformation from the m-phase to the t-phase occurred in the powder, with Ce^4+^ partially entering the Zr lattice to form the t-Zr_0.84_Ce_0.16_O_2_ phase, which facilitated the suppression of the m-phase and improved the high-temperature phase stability. It was also found that the PS-PVD coatings prepared using the aforementioned powders exhibited coarser columnar structures with increasing powder calcination temperature.

## 1. Introduction

Plasma spray–physical vapor deposition (PS-PVD) is an emerging coating deposition technology, which has evolved from low-pressure plasma spray–thin film (LPPS-TF) technology. The emergence of this technology fills the technical gap between Atmospheric Plasma Spray (APS) and Electron Beam–Physical Vapor Deposition (EB-PVD), combining the advantages of both [1]. PS-PVD allows for the selective deposition of porous, highly insulating columnar coatings, and dense, uniformly layered coatings [2]. Traditional plasma spray coatings are formed by the deformation, spreading, and stacking of molten powder particles upon impact. In contrast, PS-PVD can achieve the single-phase or multiphase deposition of gas, gas–liquid, or gas–liquid–solid according to the requirements of the coating structure through parameter control, to obtain the corresponding structured coatings. PS-PVD technology enables the gasification and vapor deposition of yttria-stabilized zirconia (YSZ) powder particles at relatively low working pressures (about 100–200 Pa) and high input powers (up to 180 kW), thereby achieving feathered columnar structure coatings [3]. This technology achieves high deposition efficiency (>10 μm·min^−1^) and enables line-of-sight deposition. The thermal barrier coatings prepared by it possess numerous pores and exhibit a typical columnar structure, resulting in low thermal conductivity (<1.5 W·m^−1^·K^−1^, 1000 °C), high stress tolerance (over 5000 cycles at 1050 °C air cooling), and comparable bond strength to EB-PVD thermal barrier coatings (>50 MPa). PS-PVD coatings have become a major focus in thermal barrier coating research. Currently, 6% to 8% partially yttria-stabilized zirconia (YSZ) is the most widely used ceramic material for practical thermal barrier coatings. It boasts good thermal conductivity, excellent thermal cycling performance, and a relatively high coefficient of thermal expansion. However, YSZ coatings exhibit poor high-temperature phase stability. At temperatures around 1200 °C, YSZ undergoes phase transitions during cooling, accompanied by significant volume changes, leading to increased and uneven internal stresses, eventually causing spalling failure [4]. Moreover, YSZ undergoes severe sintering densification at high temperatures, inevitably compromising the thermal and mechanical properties of the coating. As the operating temperatures of advanced aerospace engines continue to rise, traditional YSZ coatings, due to their inherent limitations, struggle to meet the more demanding service environments and fail to provide adequate protection for hot section components. Therefore, improving or developing thermal barrier coating materials with excellent thermal stability at high temperatures is a key research priority. CeO_2_ is one of the most common rare earth oxides used for doping-modified YSZ coatings. It has a cubic crystal structure and limited solid solubility in ZrO_2_. Relevant studies have shown that ZrO_2_ coatings co-stabilized with CeO_2_ and Y_2_O_3_ exhibit better high-temperature phase stability, superior resistance to molten salt corrosion, and improved thermal insulation, thermal expansion coefficient, and thermal shock resistance due to the doping of CeO_2_ [5,6,7].

Although PS-PVD technology has many advantages in preparing thermal barrier coatings, it demands the high gasification and flowability of the sprayed powder. In order to facilitate the gasification of the sprayed powder in the plasma jet, nanometer-grade powder is often selected as the PS-PVD spray powder [8]. Due to the small particle size and large specific surface area of nanometer-grade powder, its flowability is poor, which easily causes blockages in the powder delivery process and is generally difficult to directly use for PS-PVD spraying coatings. The particle size of the spraying powder affects its vaporization and fluidity, which in turn affects whether the spraying can be carried out smoothly or not, so choosing the right spraying powder is one of the key issues in PS-PVD technology. To address this issue, people often mix nanometer powder with organic dispersants and binders to prepare agglomerated nanometer powders by spray-drying. The presence of organic substances ensures the cohesion of the agglomerated powder particles, and the powder’s sphericity is relatively good. This type of agglomerated powder retains the gasification characteristics of nanometer-grade powder and also has good flowability due to the increased particle size. However, the presence of organic matter can affect the purity of agglomerated powders. In addition, these organic substances are susceptible to environmental changes, which are not conducive to the long-term preservation of agglomerated powders, and in the actual PS-PVD spraying process, the presence of organic substances may interfere with the spraying process, or even make it difficult to carry out the spraying process. In order to obtain a less impure powder suitable for PS-PVD spraying, the agglomerated powder needs to be calcined to decompose or volatilize the organic substances. As the calcination process proceeds, the internal structure of the agglomerated powder changes. The decomposition or volatilization of organic substances reduces the bonding force between the agglomerated powder particles, making them loose and fragile, and also making it difficult to maintain their good spherical shape. Therefore, an appropriate calcination process is needed to ensure that there is a certain degree of sintering between nanometer particles to provide a new source of cohesion for agglomerated powder particles. However, an excessive calcination temperature or prolonged calcination time can lead to excessive sintering between the nanometer particles within the agglomerated powder particles, resulting in a decrease in the gasification performance of the agglomerated powder. Therefore, a suitable calcination process for agglomerated powders is crucial for the flow properties and vaporization properties of sprayed powders.

This study primarily investigates the influence of the calcination temperature on the microstructure, composition, and properties of CeO_2_-Y_2_O_3_-ZrO_2_ (CYSZ) agglomerated powder. It also utilizes the calcined powder to study the microstructure and properties of corresponding coatings prepared by the PS-PVD method.

## 2. Materials and Methods

### 2.1. Materials

Nanopowders of 8YSZ prepared by the electric fusion method (Zhengzhou Zhenzhong Fused New Material Co. Ltd., Zheng Zhou, China) and CeO_2_ powder prepared by the chemical precipitation method (Shanghai ST-nanoscience & technology Co., Ltd., Shanghai, China) were selected as raw materials. The average particle size of the 8YSZ powder is approximately 50 nm, while the average particle size of the CeO_2_ powder is 2 μm.

### 2.2. Preparation of Agglomerated Powders

The mass ratio of 8YSZ to CeO_2_ nanopowders was 10:1. The mixed powder, zirconia ball milling beads, and deionized water were put into a zirconia ball milling jar at a mass ratio of 1:2:1, and then milled and mixed for 20–25 h at 350–400 rpm using a planetary ball mill (ND7-04, Nanda Tianzun Electronics Co., Ltd., Nanjing, China). After milling and mixing, a certain proportion of dispersant PAA and binder PVP were added sequentially, and the mixture was milled for 180 min and 120 min, respectively, to obtain a suspension of mixed powders suitable for spray granulation. The composition of the dispersant and binder in the suspension is shown in Table 1.

CYSZ agglomerated powder was prepared using a mobile spray-dryer (Mobile Minor, Gea Niro, Müllheim, Germany), and the schematic diagram of the spray-drying process is shown in Figure 1. The suspension was pumped into the atomizing nozzle by a feed pump, where it was atomized into droplets with an average diameter of 20–60 μm under the action of high-speed compressed air flow. The droplets, under the influence of surface tension, contracted into spherical shapes. When exposed to the hot air in the drying chamber, the moisture on the surface of the atomized droplets evaporated rapidly. The evaporation interface moved towards the center of the droplets, and as the moisture evaporated, the droplets shrank and dried into spherical powders. Under the negative pressure of the exhaust fan, the mixed airflow of powder and air was drawn into a cyclone separator, where the powder and air were separated due to centrifugal force. The exhaust gas was discharged from the induced draft fan through a bag filter, while the powder was collected from the bottom of the cyclone separator. The specific process parameters of the spray-drying are shown in Table 2.

### 2.3. Agglomerated Powder Calcination

Before calcination of the agglomerated powder, thermal gravimetric analysis (TG) of the spray-dried agglomerated powder was conducted from room temperature to 1300 °C using an STA 409 PC Luxx thermogravimetric analyzer (Netzsch, Hanau, Germany) at a heating rate of 20 °C/min. The TG curve of the spray-dried powder is shown in Figure 2. It can be observed that there is a significant endothermic peak and a sharp mass loss process in the temperature range of 300 °C to 500 °C, which is mainly due to the volatilization of the dispersant PAA and the binder PVP in the powder [9]. To ensure complete decomposition of organic matter in the agglomerated powder, the calcination temperature should be set above 500 °C. Therefore, the spray-dried powder was calcined at 600 °C, 700 °C, 800 °C, and 900 °C with a heating rate of 5 °C/min and a holding time of 3 h. This study will investigate the powders calcined at four different temperatures and the agglomerated powders without calcination treatment after spray-drying, as shown in Table 3.

### 2.4. Coating Preparation

A 100–120 μm thick MCrAlY bond coating and a 200–220 μm thick CYSZ coating were prepared on K4169 high-temperature alloy discs using PS-PVD (Oerlikon-Metco, Wohlen, Switzerland) spraying equipment, with a high-power single-cathode O3CP spray gun selected for spraying. The deposition parameters for the CYSZ coating are shown in Table 4.

### 2.5. Characterization of Powders and Coatings

Particle size distribution and specific surface area of the powder were measured using a laser diffraction particle size analyzer (Mastersizer-3000, Malvern, UK). Due to the poor flowability of the nano-agglomerated powder, it was challenging to measure using a Hall flowmeter. Therefore, a powder characterization tester (Keyence BT-1000, Shanghai, China) was used to measure the Hausner ratio indirectly, reflecting the flowability of the nano-agglomerated powder. The microstructures of the powders and coatings prepared by PS-PVD were characterized using scanning electron microscopy (SEM, Nova Nano-450, FEI, Leiden, The Netherlands). Powder and coating analysis was conducted using an X-ray diffractometer (XRD, Rigaku Smartlab 9 kW, Akishima, Japan) with a scanning angle range of 10°–90°, a scanning step size of 0.02°/s, and a Cu Kα radiation source. The bond strength of the coatings was determined according to HB-5476 standards [10]. Tensile strength tests were conducted using a universal tensile testing machine (gp-ts200m).

## 3. Results and Discussion

### 3.1. Effect of Calcination Temperature on the Microstructure of the Powders

Figure 3(a1–e3) show the spray-dried powder (1#) and the agglomerated powders subjected to different temperature calcination treatments (2–5#). From Figure 3(a1,a2), it can be seen that the agglomerated powders obtained through spray-drying exhibit differences in size, but most of them have good sphericity. Among them, the powders with excessively large or small particle sizes are not suitable for PS-PVD spraying and can be screened out to avoid powder blockage and difficulty in powder gasification. Figure 3(a3) shows a magnified image of the agglomerated powder, from which it can be observed that the powder particles are agglomerated from nano-sized YSZ powder and larger micron-sized CeO_2_, with a surface covered by a considerable amount of organic residues. These organic residues are the main guarantee for maintaining the spherical particle state of the agglomerated powder obtained by spray-drying.

Figure 3(b1–b3) shows the powder calcined at 600 °C (2#). Compared to the spray-dried powder, it can be observed that there is less nano-sized YSZ powder on the particle surface, and the organic residues are lost due to volatilization during calcination. This loss of organic material diminishes the binding capability between the powder particles. Additionally, due to the relatively low calcination temperature, sintering between particles is difficult, which prevents the formation of new sources of cohesion. As a result, the agglomerated powder particles become loose, with internal powder separation tendencies, and the spherical state of the particles is challenging to maintain, leading to the formation of powder agglomerates.

Figure 3(c1–c3) depicts the morphology of the powder calcined at 700 °C (3#). Compared to the powder calcined at 600 °C (2#), it can be observed that the powder calcined at 700 °C (3#) exhibits higher sphericity and larger particle size. A large amount of nano-sized YSZ is attached to the spherical particles, and the surface of the powder particles is denser. This may be attributed to some nano-particles undergoing slight sintering as the calcination temperature increases. With the volatilization of organic material, new binding forces are provided to maintain a higher sphericity of the agglomerated powder. However, at this stage, the surface of the agglomerated powder particles is still not smooth enough, and there are more large pores, resulting in a larger specific surface area and poorer overall flowability of the powder particles.

When the calcination temperature reaches 800 °C (4#), the sphericity of the powder (4#) is further improved. The magnified view of the particle surface in Figure 3(d3) shows that compared to the 3# powder, the 4# powder exhibits further slight sintering between the nano-sized particles, resulting in a denser and smoother particle surface with fewer large pores. This leads to a reduction in the specific surface area of the agglomerated powder, which is more conducive to enhancing the flowability of the powder.

Raising the calcination temperature to 900 °C, the surface of the agglomerated powder (5#) becomes denser and smoother. However, as seen in the magnified view in Figure 3(e3), the agglomerated powder particles show more pronounced sintering, with the original fine powders combining into larger block-like particles. While the smoother and denser surface of the agglomerated powder improves its flowability, excessive sintering may lead to a decrease in the powder’s gasification performance.

In summary, the calcination of the agglomerated powders obtained through spray-drying effectively removes organic residues and different calcination temperatures affect the microstructure of the agglomerated powder particles. At lower calcination temperatures, the volatilization of organic residues, which are crucial for maintaining the spherical shape of the agglomerated powder particles, leads to a porous and less spherical morphology due to insufficient internal sintering to provide new binding forces. As the calcination temperature increases, further internal sintering occurs between the fine powders, enhancing the cohesion between particles, improving sphericity, smoothing the surface, and reducing the number of pores. However, at 900 °C, the mutual sintering between the nano-particles becomes more severe, gradually forming more localized block-like structures.

### 3.2. Influence of Calcination Temperature on Particle Size Distribution and Flowability

Figure 4 shows the particle size distribution curves of the spray-dried powder (1#) and the calcined powders (2–5#), while Figure 5 displays the corresponding particle sizes for different powders with 10%, 50%, and 90% of the cumulative distribution, respectively. As shown in Figure 4, the particle size of the spray-dried powder is more concentrated, with a shorter distribution range and a higher peak compared to the calcined powders. Figure 5 indicates that the particle size of the spray-dried powder is generally smaller, with a D90 of only 31.2 μm, which is much smaller than the D90 particle size of the calcined powders. This is because the spray nozzle of the spray granulator limits the size of the spray droplets within a certain range.

The particle size distribution of the powders undergoes significant changes after calcination treatment. Particularly, the powder calcined at 600 °C (2#) exhibits a distinctive size distribution. As observed from Figure 4 and Figure 5, the particle size range of the 2# powder is broader, with a lower curve peak and D10 and D50 values of 4.2 μm and 13.2 μm, respectively, both lower than the pre-calcination sizes. This is because the lower calcination temperature results in the volatilization of organic compounds without sufficient sintering between particles, failing to provide new sources of cohesion, ultimately causing the loose agglomerated powder particles to split into many smaller ones.

The particle size distribution curves of the agglomerated powders calcined at temperatures of 700 °C, 800 °C, and above 900 °C are shown in Figure 4. From the graph, it can be observed that none of the three types of agglomerated powders exhibit a phenomenon similar to that of powder 2#, which had a significant amount of small-sized particles. This indicates that internal sintering occurred within these agglomerated powders, providing internal cohesion and preventing the loose splitting of the agglomerated powder particles. However, the distribution of the large-sized particles in powders 3#, 4#, and 5# is more prominent, with D50 and D90 values greater than those of the non-calcined powders. This is because during calcination, organic compounds decompose first, causing the agglomerated powder to loosen and expand. Subsequently, with continued calcination, internal sintering gradually occurs, maintaining the expansion of the agglomerated powder particles. The preparation process of the suspension slurry and the selection of organic dispersants/binders may also contribute to the increase in the large-sized particle powders. As the calcination temperature increases, the sintering shrinkage of powder particles strengthens, leading to a gradual reduction in the particle size.

Figure 6 shows the specific surface area of the spray-dried powder (1#) and the calcined powders (2–5#). The specific surface area is one of the parameters affecting the powder flowability. Figure 7 displays the tapped density, bulk density, and Hausner ratio of the aforementioned powders. The Hausner ratio is another parameter of the powder fluidity, which is the ratio of the tap density of a powder to its bulk density. When calcined at 600 °C, the powder particles exhibit the maximum specific surface area, attributed to the loose agglomeration of powder particles and the presence of numerous surface voids. Moreover, the increase in the tapped density results in a high Hausner ratio of up to 1.52 for powder 2#, indicating poor flowability. As the calcination temperature increases from 700 °C to 900 °C, there is a slight decrease in the specific surface area and a slight increase in the Hausner ratio, indicating overall better flowability compared to powder 2#.

### 3.3. Effect of Calcination Temperature on Composition of the Powders

The traditional 6~8% YSZ exhibits different crystalline structures at various temperatures, including the monoclinic phase (m-phase), tetragonal phase (t-phase), cubic phase (c-phase), and metastable tetragonal phase (t’-phase) between the t-phase and c-phase [11]. In the c-phase of ZrO_2_, there are eight oxygen ions distributed around each Zr^4+^ ion. When the oxygen ions are stretched along the c-axis direction, the lattice becomes more tetragonal, forming the t-phase. The t’-phase, located between the c-phase and t-phase, experiences less stretching of the oxygen ions along the c-axis direction [12]. Pure ZrO_2_ without doping undergoes phase transitions with significant volume changes under temperature variations. Heating (around 1200 °C) induces a transformation from the m-phase to the t-phase, resulting in volume contraction, while cooling (around 1000 °C) leads to a transformation from the t-phase to the m-phase, causing volume expansion. Therefore, pure ZrO_2_ is unsuitable for thermal barrier coatings [13]. Doping with an appropriate amount of Y_2_O_3_ can partially stabilize ZrO_2_. At high temperatures, the t-phase can coexist with the c-phase or even entirely dominate as the t-phase, while after cooling, it can be preserved as the metastable t’-phase, avoiding the volume changes associated with the t→m-phase transition and the resulting internal stresses [14]. However, when the temperature reaches 1200 °C, the Y^3+^ in YSZ tends to segregate and enrich in the c-phase, making it difficult for the t’-phase to persist during cooling. It decomposes into the c-phase and t-phase, ultimately leading to the transformation of the t-phase into the m-phase and causing volume expansion [15]. CeO_2_ has a cubic crystal structure and can be limitedly solid-soluted into ZrO_2_, with a large solid solution range. It can enhance the high-temperature phase stability, thermal expansion coefficient, thermal cycling performance, and thermal insulation properties of YSZ, making it a commonly used doping-modified rare earth oxide in YSZ [16]. Doping with Ce^+4^ with a larger atomic radius increases the crowding of the coordinated oxygen ions in the t-ZrO_2_ phase, further exacerbating the misfit dislocation deformation of the coordinated oxygen ions along the c-axis direction. The stretching deformation of the oxygen atom dislocations inevitably leads to changes in the Zr atom distances and an increase in the tetragonality. The significant differences in the atomic radius and mass between the dopant and Zr create a strong localized stress field, increasing the potential energy barrier for the t→m-phase transition and inhibiting the formation of the m-phase [17].

The XRD patterns of the five powders (1–5#) are shown in Figure 8. It can be observed from the figure that the spray-dried powder is mainly composed of the t-phase with a small amount of the m-phase. With the progression of the calcination process and the increase in the calcination temperature, the small amount of m-phase gradually transforms into the t-phase, resulting in a decrease in the m-phase content. Additionally, a portion of Ce^4+^ enters the Zr lattice, forming the t-Zr_0.84_Ce_0.16_O_2_ phase. These results indicate that the calcination treatment of CYSZ spray-dried powder can improve the phase stability of agglomerated powder.

### 3.4. Effect of Powder Variation on the Microstructure, Composition, and Properties of the Prepared Coatings

Figure 9 depicts the surface and cross-sectional microstructures of the PS-PVD CYSZ coatings prepared from calcined powders (2–5#). It can be observed that the PS-PVD coatings produced from the powders calcined at different temperatures exhibit typical feather-like structures. However, as the calcination temperature of the powder increases, the feather-like structures become coarser, with larger gaps between columns, and the cauliflower-like structures on the coating surface also increase in size. This may be attributed to the intensified sintering between secondary particles within the agglomerated powder particles as the calcination temperature rises, leading to a decrease in the powder’s gasification performance and consequently affecting the feather-like structure of the coating [18,19].

The tensile bond strength of the PS-PVD coatings prepared from calcined powders was tested, and the results are shown in Figure 10. The results indicate that the coatings have a relatively high bond strength. Specifically, the tensile strengths of the coatings prepared from powder 2# and 3# reached 75 MPa and 78 MPa, respectively. However, the tensile strengths of the coatings prepared from powder 4# and 5# slightly decreased compared to the former. This may be attributed to the microstructure of the coatings.

## 4. Conclusions

This study prepared CeO_2_-doped modified CYSZ powders and coatings. By pre-treating CYSZ spray-dried powders used for PS-PVD spraying through calcination, the spray performance, phase composition, and coatings of different calcination temperatures for sprayable CYSZ powders were comparatively analyzed. The results indicate that the calcination pre-treatment of the spray-dried CYSZ powders for PS-PVD significantly influences their spray performance, phase composition, and the microstructure of the prepared coatings.

The agglomerated powders calcined at 600 °C exhibited poor flowability due to the mild sintering between the secondary particles within the powder grains, resulting in weak internal cohesion and easy fragmentation. Compared to the former, the agglomerated powders calcined at 700 °C showed improved flowability as the sintering between the powder particles inside is more thorough, reducing the likelihood of fragmentation. The coatings prepared via PS-PVD from these powders exhibit a typical feather-like microstructure and better overall performance. However, the agglomerated powder calcined at 800 °C and 900 °C, while possessing sufficient sintering for internal cohesion, suffers from reduced gasification performance due to excessive sintering, resulting in coarser feather-like structures in the prepared coatings. In summary, calcination at 700 °C is deemed the most suitable pre-treatment temperature for spray-dried powders used in PS-PVD. The agglomerated powders calcined at this temperature exhibit superior spray performance, phase stability, and microstructure, making them excellent candidates for PS-PVD spray powders.

## Figures and Tables

**Figure 1 nanomaterials-14-00995-f001:**
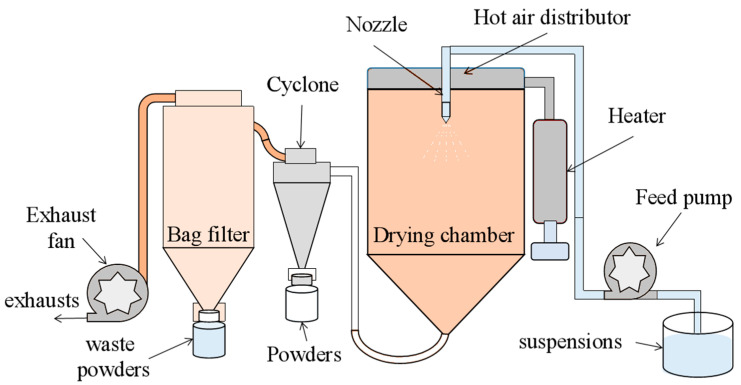
Schematic diagram of spray-drying process.

**Figure 2 nanomaterials-14-00995-f002:**
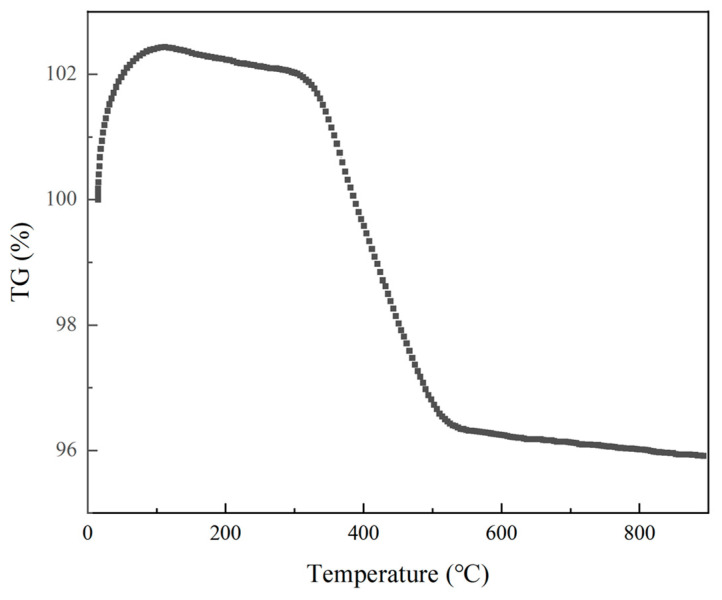
TG curves of spray-dried powders.

**Figure 3 nanomaterials-14-00995-f003:**
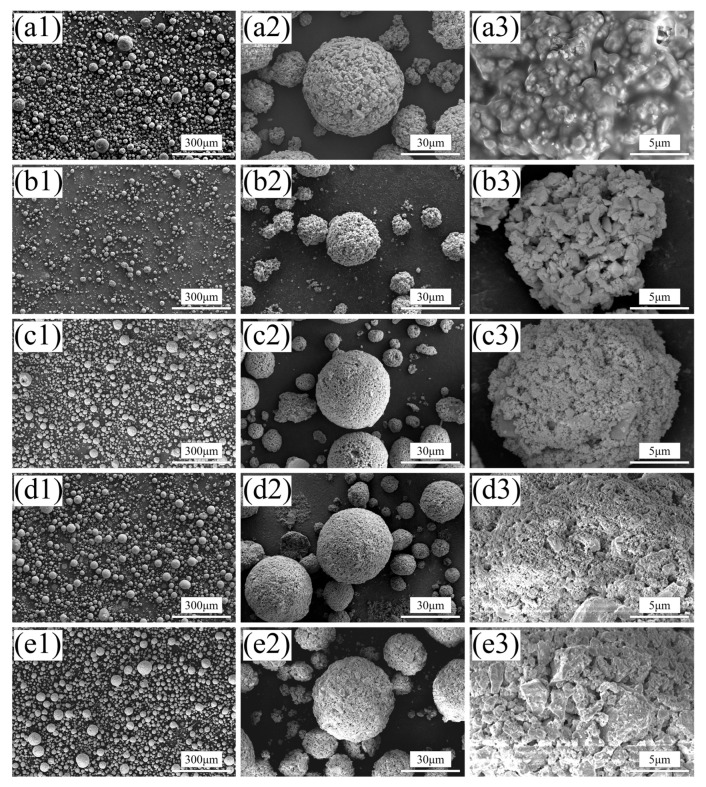
SEM morphology of the spray-dried and calcined powders, (**a1**–**a3**) powder 1# (spray-dried), (**b1**–**b3**) powder 2# (calcined at 600 °C), (**c1**–**c3**) powder 3# (calcined at 700 °C), (**d1**–**d3**) powder 4# (calcined at 800 °C), (**e1**–**e3**) powder 5# (calcined at 900 °C).

**Figure 4 nanomaterials-14-00995-f004:**
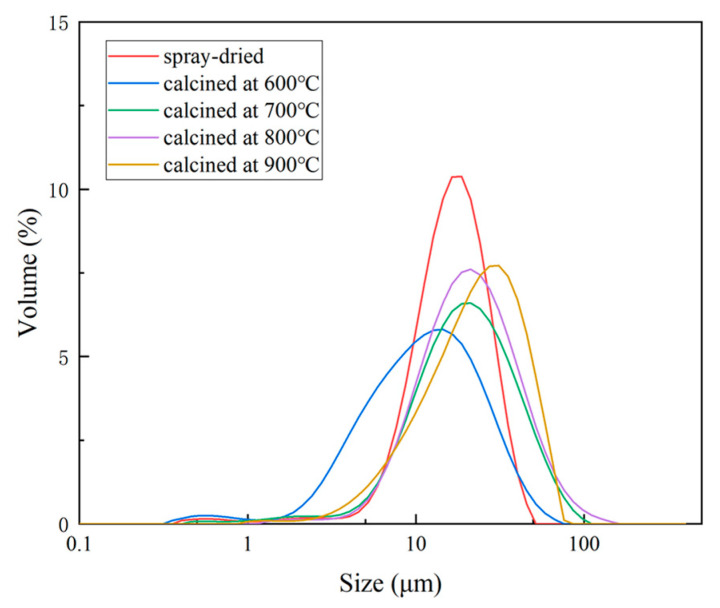
Particle size distribution curve of the spray-dried and calcined powders.

**Figure 5 nanomaterials-14-00995-f005:**
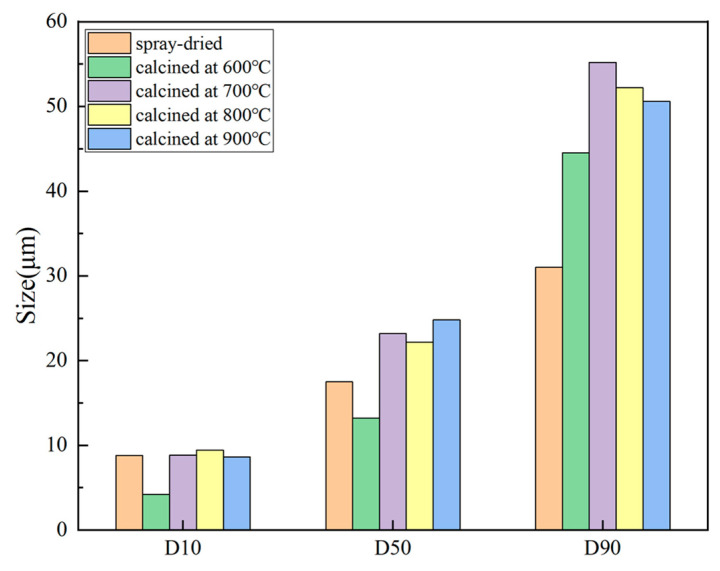
D10, D50, and D90 particle sizes of the spray-dried and calcined powders.

**Figure 6 nanomaterials-14-00995-f006:**
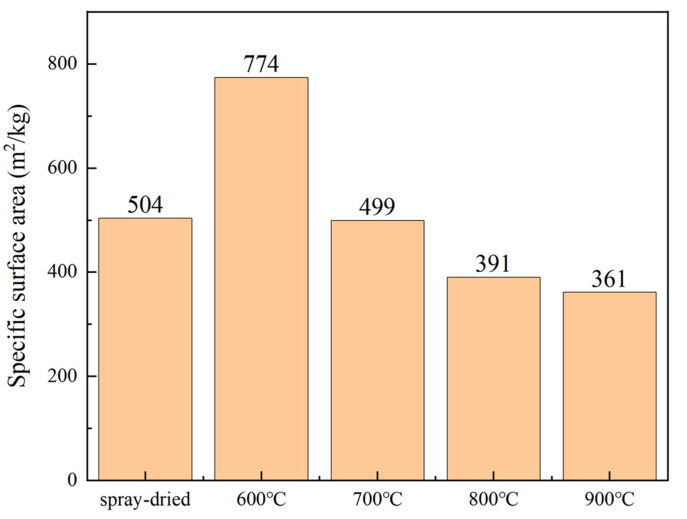
Specific surface area of the spray-dried and calcined powders.

**Figure 7 nanomaterials-14-00995-f007:**
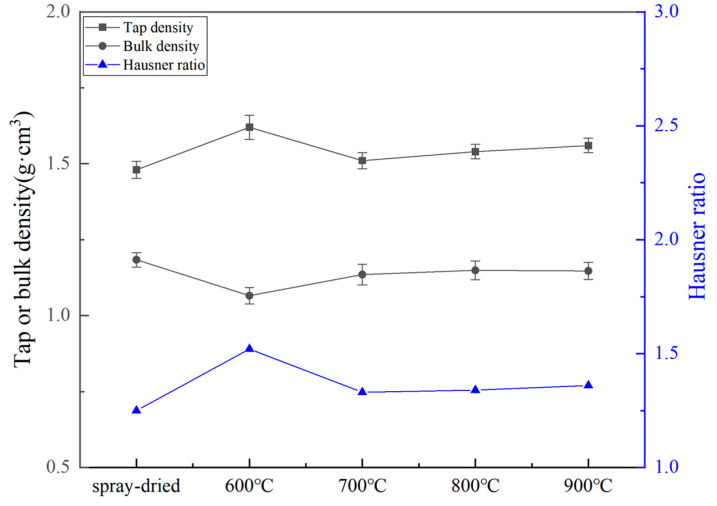
Tap density, bulk density, and Hausner ratio of the spray-dried and calcined powders.

**Figure 8 nanomaterials-14-00995-f008:**
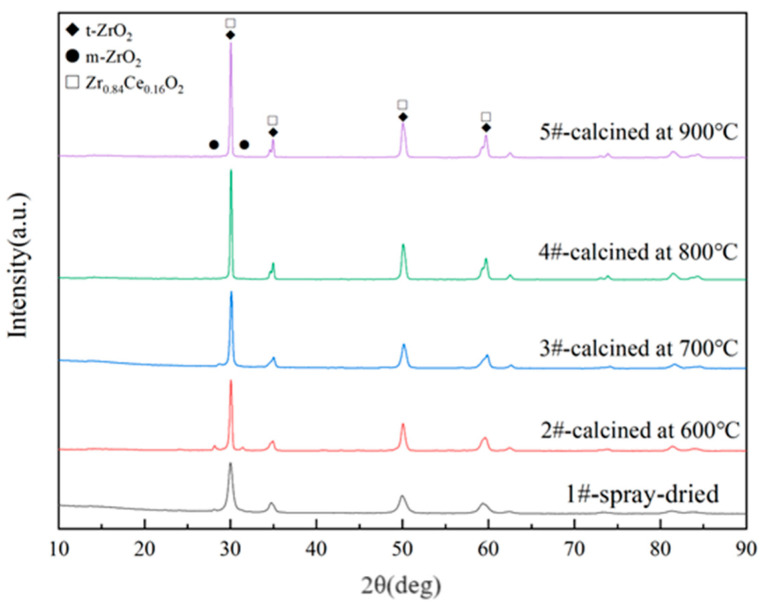
XRD profiles of the spray-dried and calcined powders.

**Figure 9 nanomaterials-14-00995-f009:**
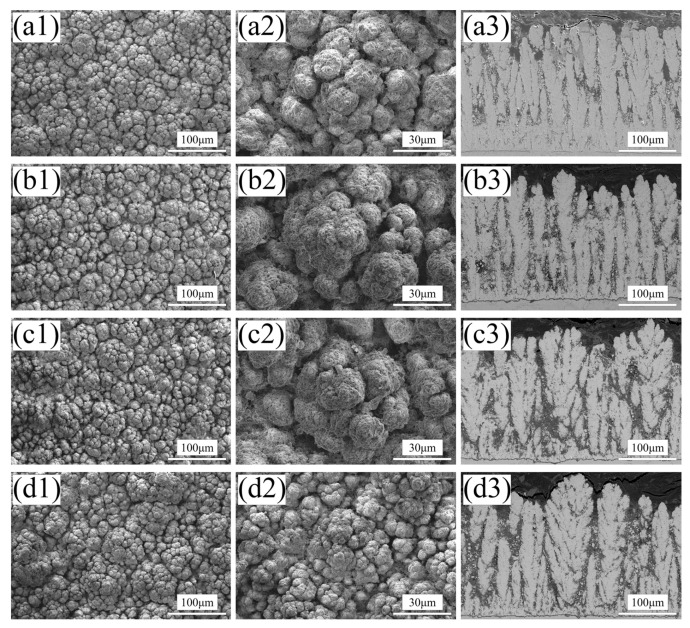
SEM micrographs of the CYSZ coatings deposited by the calcined powders, (**a1**–**a3**) powder 2# (calcined at 600 °C), (**b1**–**b3**) powder 3# (calcined at 700 °C), (**c1**–**c3**) powder 4# (calcined at 800 °C), (**d1**–**d3**) powder 5# (calcined at 900 °C).

**Figure 10 nanomaterials-14-00995-f010:**
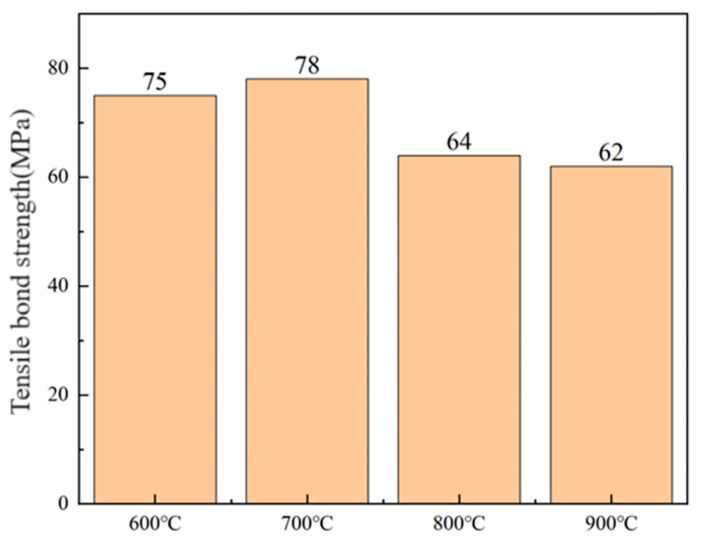
Tensile bond strength of the coatings prepared by the spray-dried and calcined powders.

**Table 1 nanomaterials-14-00995-t001:** List of organic ingredients used in spray-dry suspension.

Type	Product	CAS	Supplier
Dispersant	Polyacrylic acid (PAA)	9003-01-4	Tianjin Damao Chemical Reagent Factory, Tianjin, China
Binder	Polyvinylpyrrolidone (PVP)	9003-39-8	Institute of New Materials, Guangdong Academy of Sciences

**Table 2 nanomaterials-14-00995-t002:** Parameters of spray-dryer.

Inter Temp./℃	Outlet Temp./°C	Feeding Speed/r·min^−1^	Pressure/Mpa
250	120	35	0.2

**Table 3 nanomaterials-14-00995-t003:** Five powder numbers.

Sample	Calcination Temp./°C	Holding Time/h	Heating Rate/°C·min^−1^
1#	Spray-dried	/	/
2#	600	3	5
3#	700	3	5
4#	800	3	5
5#	900	3	5

**Table 4 nanomaterials-14-00995-t004:** Deposition parameters of CYSZ coatings.

Ar (L/min)	He (L/min)	Current (A)	Stand-off Distance (mm)
30–40	55–65	2500–2600	900–1000

## Data Availability

The original contributions presented in the study are included in the article, further inquiries can be directed to the corresponding authors.

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
