# Peer review of "Effect of Calcination Temperature on the Microstructure, Composition and Properties of Agglomerated Nanometer CeO2-Y2O3-ZrO2 Powders for Plasma Spray–Physical Vapor Deposition (PS-PVD) and Coatings Thereof"

_nanomaterials, 2024, doi:10.3390/nano14120995_

Round 1
Reviewer 1 Report
Comments and Suggestions for Authors
The manuscript reprts a process termed "CYSZ powders for plasma spray-physical vapor deposition (PS-PVD) and coatings thereof".
I have a few relatively minor comments, explained below.
Page 3, line 106
What material pots and bowls were used? There is no description.
Page 5
What equipment was used to measure the specific surface area? Not stated. There is no explanation such as BET method was used.
Addressing these concerns would strengthen the conclesions of the manuscript.
Comments on the Quality of English Language
The manuscript reprts a process termed "CYSZ powders for plasma spray-physical vapor deposition (PS-PVD) and coatings thereof".
I have a few relatively minor comments, explained below.
Page 3, line 106
What material pots and bowls were used? There is no description.
Page 5
What equipment was used to measure the specific surface area? Not stated. There is no explanation such as BET method was used.
Addressing these concerns would strengthen the conclesions of the manuscript.
Reviewer 2 Report
Comments and Suggestions for Authors
Well written manuscript about of agglomerated CYSZ nanopowders for plasma spray-physical vapor deposition (PS-PVD) and coatings. Present interesting insight on effects that calcination temperatures have on powders properties and structure of coatings. However, there are still few thing to be added in order to improve the quality of the manuscript. In addition Title need to be corrected in order to evade misleading of readers about its content, since only nanosize is those of CYSZ particles while powders have microsize. More detailed corrections and suggestions are in the PDF.

Reviewer 3 Report
Comments and Suggestions for Authors
The article studies the influence of calcination temperature on the microstructure, composition, and properties of CYSZ aggregate powder and the properties of corresponding coatings prepared by the PS-PVD method.
There are the following comments to the article.
1. The heading to table 3 says “Four powder numbers”, but the table lists 5 powders.
2. In line 164, the section numbering should be 3, but not 3.3.
3. Judging by the Hausner ratio, spray-dried powder has the best flowability. In addition, this powder has the same specific surface area as the powder calcined at 700 °C. Again, in Fig. 8 the XRD profiles of spray-dried powder and of that calcined at 700 °C are practically the same. The question arises why section 3.4 does not contain information about the properties of the coating obtained from spray-dried powder. It is reasonable to assume that the properties of coatings obtained from non-calcined powder may be the best. The authors would need to explain why the properties of the mentioned coatings are missing, or insert the missing data. Otherwise, the article looks like an unfinished work.
Reviewer 4 Report
Comments and Suggestions for Authors
The manuscript, “Effect of calcination temperature on the microstructure, composition and properties of nanometer agglomerated CYSZ powders for plasma spray-physical vapor deposition (PS-PVD) and coatings thereof”, submitted to Nanomaterials by Hou et al. presents a study on the synthesis and characterisation of ceria-yttria-zirconia (CYSZ) mixed oxide powders, which find applications as thermal-barrier coatings, e.g., in the aerospace engineering applications.
The authors have prepared deposition powders, based on commercial YSZ (8% yttria) powder, to which 9 wt.% (10:1 ratio) ceria was added, along with PAA/PVP dispersant/binder via ball-milling, followed by spray-drying to obtain an aggregate powder, which was used either as-is, or subjected to calcination for 3h at 600 - 900 °C, with the specific range determined to yield a complete vaporisation of the polymer binder via TGA analysis. The powders were characterised for crystallinity, particle size distribution and morphology via XRD, laser scattering, and SEM. Coatings of these powders were deposited via PS-PVD and results on effects of the calcination temperature on their morphology and tensile bond strength are presented. The authors conclude that a calcination at 700 °C yields the optimal powder for PS-PVD both in terms of powder rheology and mechanical properties of the resulting coating.
Overall, I am pleased with the quality of the manuscript. The authors demonstrate both a good experimental planning, a clear research idea, and a high-level expertise in the manuscript topic. There is novelty, given the use of PS-PVD. The quality of the English presentation is very good. I would recommend the manuscript of publication in Nanomaterials, wholeheartedly, however, I can see that there are some ways in which it should be improved both to allow for a higher accessibility and readability of the text. Additionally, I sense that some important details are currently not mentioned or described sufficiently in the text.
Here are some points that could be used during the revision:
(1) The abstract contains some abbreviations that are not introduced (CYSZ, PS-PVD, line 13). Same applies for YSZ and CYSZ in the Introduction.
(2) As some of the starting materials used in the paper are highly specialised, and an unfamiliar reader might not be familiar with them, I suggest that the authors add a mention about the initial composition, based on the manufacturer’s specifications. Namely - 8YSZ (YSZ-8) obtained commercially. Please add a mention that it is zirconia, stabilised by 8% yttria. This information could be added between lines 48-50, that while 6-8% yttria YSZ is widely used, 8YSZ (YSZ-8) is one of the more popular choices, to help clarify its mention in the Materials section in lines 99-103.
(3) [Optional] I don’t see a reason why the spray-dryer parameters need to be listed in a table, given that they are not varied (Table 2), they could have just been mentioned in the text. I, however, would not insist on any alterations given this comment.
(4) A major point that I see missing from the manuscript is the complete lack of mention the substrate used. There also some lacking information, that should probably be added: the substrate preparation; the materials a procedure used for the MCrAlY bond-coat, etc. The text should be descriptive enough to allow for a reproduction. I understand that this is not the main focus of the paper, and that the use of a bond coat makes the choice of substrate arbitrary, but still it is a good practice to add as much procedural details as possible in a research paper. Probably the duration of the deposition and the resulting thickness of the coatings should also be mentioned.
(5) In subsection 3.2. the authors are using some characterisation parameters that are highly-specific and since one could assume that the readership of Nanomaterials cover a wide spectrum of fields, related to materials & nano science, this could lead to confusion. I suggest that the D10, D50, and D90 parameters are briefly introduced (e.g., “the mean particle size covering 10%, 50%, and 90% of the particle-size distribution - D10, D50, and D90, respectively”), as well as the Hausner ratio (the ρtap/ρbulk density ratio, indicative of the powder flowability, when the particle distribution is also taken into account). This would improve the general accessibility of the text. A brief mention would be sufficient, without extra details, since the text needs not to be turned into a powder rheology textbook.
(6) The entire paragraph between lines 274-303 on the zirconia-yttria stabilisation seems more appropriate to the introduction section.
(7) The x-axis in Figure 8 is labelled as “2ν/deg”, which shows two inconsistencies - (i) it should be 2θ/deg; (ii) the authors denote units in brackets in most figures, as in “2θ (degrees)”
(8) Figure 10 y-axis needs to be re-labeled as “Tensile bond strength” for consistency and to avoid confusion.
Round 2
Reviewer 3 Report
Comments and Suggestions for Authors
There are no comments. The revised article can be published.
Author Response
Dear reviewer,
I am writing this letter to express my deepest gratitude to you for the time and effort you have spent on my article [title of article]. Your expert advice not only helped to refine my work, but also had a profound impact on my entire research process.
I appreciate your attention to detail and passion for improving the quality of this paper. Your constructive feedback has enabled me to gain a deeper understanding and appreciation of my research findings.
Thank you again for your generosity in sharing your knowledge and experience and for contributing to the academic community. I hope to have the opportunity to receive your guidance and advice again in the future.
Best wishes,
Zhenning Hou